# Sensitivity analysis for mistakenly adjusting for mediators in estimating total effect in observational studies

Tingting Wang,[1,2] Hongkai Li,[1,2] Ping Su,[1,2] Yuanyuan Yu,[1,2] Xiaoru Sun,[1,2] Yi Liu,[1,2] Zhongshang Yuan,[1,2] Fuzhong Xue[1,2]

TW and HL contributed equally.

[1]Department of Biostatistics, School of Public Health, Shandong University, Jinan, China
[2]Cheeloo Research Center for Biomedical Big Data, Shandong University, Jinan, China

**Correspondence to**
Professor Fuzhong Xue;
xuefzh@sdu.edu.cn

## ABSTRACT

**Objectives** In observational studies, epidemiologists often attempt to estimate the total effect of an exposure on an outcome of interest. However, when the underlying diagram is unknown and limited knowledge is available, dissecting bias performances is essential to estimating the total effect of an exposure on an outcome when mistakenly adjusting for mediators under logistic regression. Through simulation, we focused on six causal diagrams concerning different roles of mediators. Sensitivity analysis was conducted to assess the bias performances of varying across exposure-mediator effects and mediator-outcome effects when adjusting for the mediator.

**Setting** Based on the causal relationships in the real world, we compared the biases of varying across the effects of exposure-mediator with those of varying across the effects of mediator-outcome when adjusting for the mediator. The magnitude of the bias was defined by the difference between the estimated effect (using logistic regression) and the total effect of the exposure on the outcome.

**Results** In four scenarios (a single mediator, two series mediators, two independent parallel mediators or two correlated parallel mediators), the biases of varying across the effects of exposure-mediator were greater than those of varying across the effects of mediator-outcome when adjusting for the mediator. In contrast, in two other scenarios (a single mediator or two independent parallel mediators in the presence of unobserved confounders), the biases of varying across the effects of exposure-mediator were less than those of varying across the effects of mediator-outcome when adjusting for the mediator.

**Conclusions** The biases were more sensitive to the variation of effects of exposure-mediator than the effects of mediator-outcome when adjusting for the mediator in the absence of unobserved confounders, while the biases were more sensitive to the variation of effects of mediator-outcome than those of exposure-mediator in the presence of an unobserved confounder.

## Strengths and limitations of this study

► For six different causal diagrams, we compared biases of distinct adjustment strategies with and without adjusting for mediators by conducting simulation studies.
► Sensitivity analysis was conducted to assess the performances of varying across the effects of exposure-mediator and mediator-outcome.
► The simulation schemes and parameters were conducted mainly based on real observational studies.
► The combination of theoretical derivation and simulation studies makes the results more credible.
► The limitation of these simulation studies was that they operated under the framework of logistic regression and therefore focused on only binary variables.

## INTRODUCTION

Estimating the total effect of the exposure (E) on the outcome (D) is a great challenge in epidemiology studies because confounders are commonly confused with mediators.[1–3] If confounders and mediators are misclassified, the ability to control confounders in the estimation of the total effect of the exposure on the outcome is hampered. In fact, various strategies are used to eliminate confounding bias in observational studies. The conventional approaches include multivariate regression, stratification, standardisation and inverse-probability weighting.[4 5] Furthermore, causal diagrams provide a formal conceptual framework for identifying and selecting confounders,[6 7] so that analysis can avoid falling into analytic pitfalls.[8] In practice, even the underlying causal diagrams and the role of covariates (mediator, confounder, collider and instrumental variable) are not completely understood, as investigators usually adjust for the covariates that are associated with the outcome and exposure.[9–12] Therefore, our paper focuses on the biases of varying across the effects of exposure-mediator (E→M) and mediator-outcome (M→D) when mistakenly adjusting for mediators under the logistic regression model.

Several causal inference studies have made considerable contributions to mediation analysis by providing definitions for direct and indirect effects that allow for the

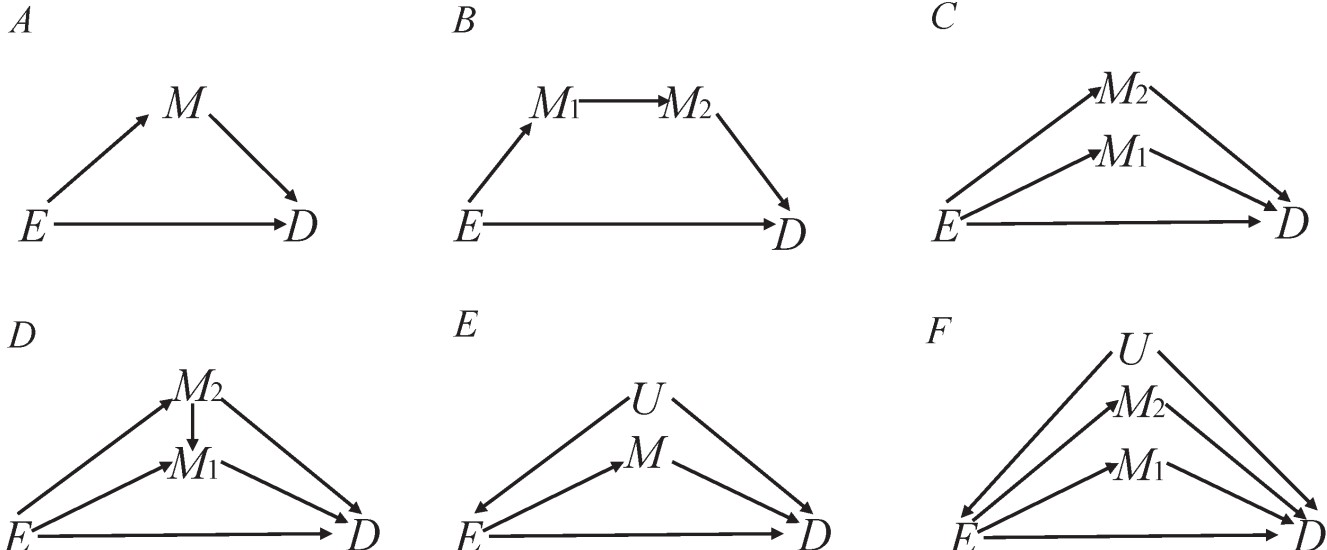

**Figure 1** Six causal diagrams were designed for estimating the causal effect of E on D. (A) a single mediator M; (B) two series mediators $M_1$ and $M_2$; (C) two independent parallel mediators $M_1$ and $M_2$; (D) two correlated parallel mediators $M_1$ and $M_2$; (E) a single mediator with an unobserved confounder U; (F) two independent parallel mediators $M_1$ and $M_2$ with an unobserved confounder U.

decomposition of a total effect into a direct and an indirect effect.[13–21] Arbitrarily adjusting for a mediator would generally bias the estimate of the total effect of the exposure on the outcome.[8 22 23] Practically, it can mistakenly identify a non-confounding risk factor as a confounder. In the perspective of causal diagrams, little attention has been paid to the biases when adjusting for mediators under the logistic regression model in estimating the total effect of E on D. Hence, we focused on the sensitivity analysis technique to assess the biases of varying across the effects of E→M and M→D when adjusting for the mediator.

In this paper, six typical causal diagrams corresponding to causal correlation are given in figure 1: a single mediator (figure 1A), two series mediators (figure 1B), two independent parallel mediators (figure 1C), two correlated parallel mediators (figure 1D), a single mediator with an unobserved confounder (figure 1E) and two parallel mediators with an unobserved confounder (figure 1F). The paper aimed to explore the sensitivity of biases to the variation of the effects of E→D and M→D when adjusting for the mediator. Hence, both theoretical proofs and quantitative simulations were performed to dissect the bias of varying across the effect of E→M and varying across the effect of M→D when adjusting for mediators under the logistic regression model.

## METHODS

A directed acyclic graph (DAG) is composed of variables (nodes) and arrows (directed edges) between nodes such that the graph is acyclic. The causal diagrams are formalised as DAGs, providing investigators with powerful tools for bias assessment.[24] It provides a device for deducing the statistical associations implied by causal relations. Furthermore, given a set of observed statistical associations, a researcher knowledgeable about causal diagrams theory can systematically characterise all causal structures compatible with the observations.[25 26]

The total effect of the exposure on the outcome can be calculated based on the *do-calculus* and *back-door* criterion proposed by Pearl.[27 28] For exposure X and outcome Y, a set of variables Z satisfies the back-door path criterion with respect to (X, Y) if no variable in Z is a descendant of X and Z blocks all back-door paths from X to Y. Then, the effect of X on Y is given by the following formula:

$$P\left(y|do\left(x\right)\right) = \sum_{Z} P\left(y|x, z\right) P\left(z\right)$$

Note that the expression on the right hand side of the equation is simply a standardised mean. The difference $E\left(Y|do\left(x'\right)\right) - E\left(Y|do\left(x''\right)\right)$ is taken as the definition of 'causal effect', where $x'$ and $x''$ are two distinct realisations of X.[23] The interventional distribution, such as that corresponding to $Y(x)$, namely, $P\left(y|do\left(x\right)\right)$, is not necessarily equal to a conditional distribution $P\left(y|x\right)$. It stands for the probability of $Y = y$ when the exposure X is set to level x. The ignorability assumption $Y\left(x\right) \perp X$ states that, if we happen to have information on the exposure variable, it does not give us any information about the outcome Y after the intervention $do\left(x\right)$ was performed. In addition, it can be shown that if ignorability holds for Y(x) and X (alternatively if there are no back-door paths from X to Y in the corresponding causal DAGs), then $P\left(y|do\left(x\right)\right) = P\left(y|x\right)$.[29 30]

Let $D_e$ and $M_e$ denote the values of the outcome and mediator that would have been observed had the exposure E been set to level e, respectively. On the OR ($OR_{E \to D}^{TE}$) scale, the total effect ($\beta_{E \to D}^{TE} = log\left(OR_{E \to D}^{TE}\right)$),

comparing exposure level e with e*, is given as the following[20 21]:

$$OR_{E \to D}^{TE} = \frac{P(D_e=1)/\{1-P(D_e=1)\}}{P(D_{e*}=1)/\{1-P(D_{e*}=1)\}}$$

While the effect $\left(\beta_{ED|M}\left(m\right)\right)$ of adjusting for mediator M by the logistic regression model can be given as the following:

$$\beta_{ED|M}\left(m\right) = \text{logit}\left\{P\left(D=1|e=1,m\right)\right\} - \text{logit}\left\{P\left(D=1|e*=0,m\right)\right\}$$

$$= \log\left\{\frac{P\left(D=1|e=1,m\right) P\left(D=0|e*=0,m\right)}{P\left(D=0|e=1,m\right) P\left(D=1|e*=0,m\right)}\right\}$$

where $P\left(D=1|e,m\right)$ denotes the probability of $D=1$ when the exposure E and mediator M have been set to level e and m, respectively. Taking figure 1A as an example, the logistic regression is as follows:

$$\text{logit}\left\{P\left(D=1|e,m\right)\right\} = \alpha_1 + \beta_0 e + \beta_2 m$$

Therefore, the total effect $\left(\beta_{E \to D}^{TE}\right)$ of exposure E on outcome D on the scale of logarithm OR was equal to

$$\beta_{E \to D}^{TE} = \log(OR_{E \to D}^{TE})$$

$$= \log\left\{\frac{P(D_e=1)/\{1-P(D_e=1)\}}{P(D_{e*}=1)/\{1-P(D_{e*}=1)\}}\right\}$$

$$= \text{logit}\left\{P(D_e=1)\right\} - \text{logit}\left\{P(D_{e*}=1)\right\}$$

$$= \text{logit}\left\{P(D_e=1|e=1)\right\} - \text{logit}\left\{P(D_e=1|e*=0)\right\}$$

$$= \text{logit}\left\{\sum_m P(D=1|e=1,m)P(m|e=1)\right\} - \left\{\sum_m P(D=1|e*=0,m)P(m|e*=0)\right\}$$

The effect estimation $(\hat{\beta}_{ED|M}(m))$ of adjusting for mediator M by the logistic regression model was equal to:

$$\hat{\beta}_{ED|M}(m) = \text{logit}\left\{\hat{P}(D=1|e=1,m)\right\} - \text{logit}\left\{\hat{P}(D=1|e*=0,m)\right\}$$

where $\hat{P}\left(D=1|e=1,m\right)$ denotes the probability of D=1 when the exposure E and mediator M have been set to level e=1 and m, respectively. Additionally, $\hat{P}\left(D=1|e*=0,m\right)$ denotes the probability of D=1 when the exposure E and mediator M have been set to level e*=0 and m, respectively. The theoretical results of other causal diagrams in figure 1 have been shown (online supplementary Appendix).

Note that the bias was defined by taking a difference between effect estimation by adjusting for the mediator using logistic regression and the total effect of exposure E on outcome D, that is, $bias = E[\hat{\beta}_{ED|M}(m)] - \beta_{E \to D}^{TE}$. We dissected the behaviour of the biases by varying across the effects of E→M and M→D when mistakenly adjusting for the mediator under the framework of the logistic regression model.

## SIMULATION

Six scenarios are designed to dissect the sensitivity of bias to the variation of the effects of exposure-mediator and mediator-outcome when adjusting for mediators under the framework of the logistic regression model; these DAGs are shown in figure 1. We made the following assumptions for the simulation: (1) all variables were binary, following a Bernoulli distribution; and (2) the effects from parent nodes to their child node were positive and log-linearly additive. Taking figure 1A as an example, we randomly generated the exposure following a Bernoulli distribution (ie, let $P(e = 1) = \pi$). Then, we used $P_M = exp(\alpha_0 + \beta_1 e) / \{1 + exp(\alpha_0 + \beta_1 e)\}$ to calculate the distribution probability of child node M from its parent node E. Similarly, $P_D = exp(\alpha_1 + \beta_0 e + \beta_2 m) / \{1 + exp(\alpha_1 + \beta_0 e + \beta_2 m)\}$ generated the distribution probability of D, where the parameters $\alpha_0$ and $\alpha_1$ denoted the intercept of M and D, respectively, and effect parameters $\beta_0$, $\beta_1$, $\beta_2$ referred to the effects of the parent node on their corresponding child node using a log OR scale.

After generating data, we dissected the behaviour of the biases between the effects of E→M and M→D when mistakenly adjusting for mediators under the logistic regression model. In scenario 1 (figure 1A), we compared performances by varying across the effects of E→M and M→D. Similarly, in scenario 2 (figure 1B), the effects of E→$M_1$, $M_1$→$M_2$ and $M_2$→D were explored. In scenario 3 (figure 1C), we dissected the effects of E→$M_1$ (E→$M_2$) and $M_1$→D ($M_2$→D). The comparison of scenario 4 (figure 1D) was the same as scenario 3 (figure 1C). In scenario 5 (figure 1E), the effects of E→M and M→D were excavated. Scenario 6 (figure 1F) was identical to scenario 3. We explored the biases when adjusting for mediators under the logistic regression model and thus identified the sensitivity of biases to the variation of the effects of exposure-mediator and mediator-outcome.

For each of the six simulation scenarios, we observed the biases of varying across distinct effects when adjusting for mediators using the logistic regression model with 1000 simulation repetitions. All simulations were conducted using software R from CRAN (http://cran.r-project.org/).

## RESULTS
### Scenario 1: one single mediator
In figure 1A, E has a direct (E→D) effect and an indirect (E→M→D) effect on D. Figure 2A depicted that the bias of varying across the effect of E→M was clearly greater than the bias of varying across the effect of M→D. That is, the sensitivity of bias to the variation of the effect E→M was greater than that of the effect of M→D when adjusting for the mediator M using the logistic regression model. In particular, if the effect of E→M was specified to zero in figure 2B, M would be associated with D conditional on E and unconditionally independent with E, and M would become an independent risk factor of the outcome, as adjusting for M would obtain a positive 'bias'. Such bias was a consequence of the non-collapsibility of the OR, and the M-conditional ORs must

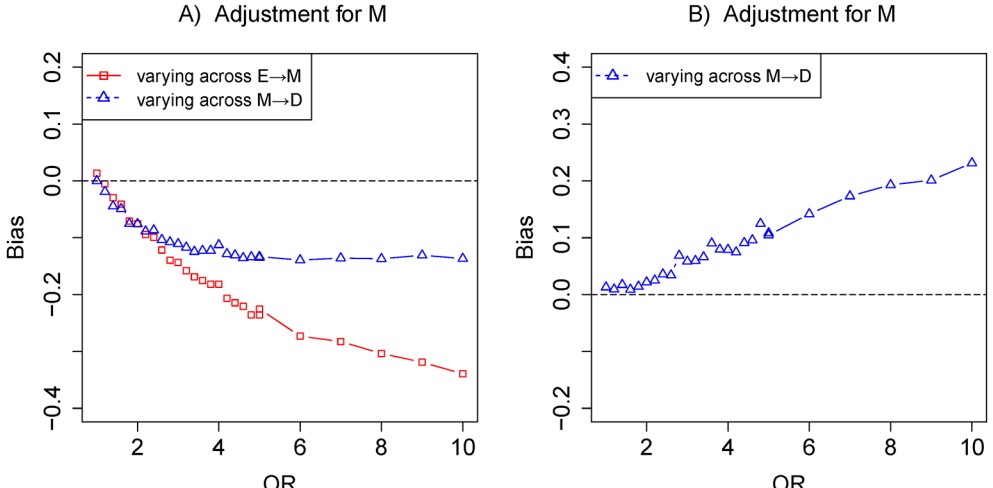

**Figure 2** The biases with the effects of E→M (red) and M→D (blue) increasing, respectively. Comparison of the bias of different effects in adjustment mediator. The OR of target effect (eg, E→M) from 1 to 10 given other effects fixed In in (A). The OR of the effect of M→D from 1 to 10 with the effect of E→M being equal to zero in (B) (colour figure online).

be farther from 1 than the unconditional ORs.[31][32] In fact, both adjustment and non-adjustment for M should yield unbiased causal effect estimates. Certainly, in this case, both the marginal OR and conditional OR obtained from standardisation and inverse-probability weighting were equal to the total effect.[33] Moreover, figure 2A indicated that adjusting for mediator M was indeed biased to the total effect of the exposure on the outcome.

The total effect $\left(\beta_{E \to D}^{TE}\right)$ of exposure E on outcome D on the log OR scale was equal to

$$\beta_{E \to D}^{TE} = log(OR_{E \to D}^{TE})$$

$$= log\left\{\frac{P(D_e = 1)/\{1 - P(D_e = 1)\}}{P(D_{e^*} = 1)/\{1 - P(D_{e^*} = 1)\}}\right\}$$

$$= log\left\{\frac{P(D_e = 1) \times \{1 - P(D_{e^*} = 1)\}}{\{1 - P(D_e = 1)\} \times P(D_{e^*} = 1)}\right\}$$

$$= log\left\{\frac{P(D = 1|e = 1) \times P(D = 0|e^* = 0)}{P(D = 0|e = 1) \times P(D = 1|e^* = 0)}\right\}$$

$$= log\left\{\frac{[\sum_m P(D = 1|e = 1, m)P(m|e = 1)] \times [\sum_m P(D = 0|e^* = 0, m)P(m|e^* = 0)]}{[\sum_m P(D = 0|e = 1, m)P(m|e = 1)] \times [\sum_m P(D = 1|e^* = 0, m)P(m|e^* = 0)]}\right\}$$

The effect $\left(\beta_{ED|M}(m)\right)$ of adjusting for mediator M by the logistic regression model can be given as follows:

$$\beta_{ED|M}(m) = logit\left\{P(D = 1|e = 1, m)\right\} - logit\left\{P(D = 1|e^* = 0, m)\right\}$$

$$= log\left\{\frac{P(D = 1|e = 1, m) \times \{1 - P(D = 1|e^* = 0, m)\}}{\{1 - P(D = 1|e = 1, m)\} \times P(D = 1|e^* = 0, m)}\right\}$$

$$= \beta_0$$

$\beta_0$ denotes coefficient of E adjusting for M using the logistic regression model. Furthermore, the effect of adjusting for M was equal to the controlled direct effect.[19] Therefore, the bias of adjusting for the mediator using the logistic regression model could be obtained that is $bias = \beta_{ED|M}(m) - \beta_{E \to D}^{TE}$. We added signs to the edges of the DAG to indicate the presence of a particular positive or

negative effect in figure 3. Therefore, we gained bias<0 under the condition of $\beta_1*\beta_2>0$ (the effect E→M $\beta_1$ and the effect M→D $\beta_2$), indicating that the total effect of E on D was biased when adjusting for M using the logistic regression model in figure 3A, B, E and F. In addition, the bias was less than zero when the effect E→M ($\beta_1$) and the effect M→D ($\beta_2$) shared same signs (ie, both the effects E→M ($\beta_1>0$) and M→D ($\beta_2>0$) were a positive sign or both the effects E→M ($\beta_1<0$) and M→D ($\beta_2<0$) were a negative sign). Furthermore, we obtained bias>0, if $\beta_1*\beta_2<0$, suggesting that the total effect of E on D was biased when adjusting for M in figure 3C, D, G and H. In addition, the bias was greater than zero when the signs of the effects E→M ($\beta_1$) and M→D ($\beta_2$) were the opposite. The results illustrated that the bias was less than zero in the case in which the effects of exposure-mediator and mediator-outcome shared the same sign; the bias was greater than zero under the circumstance in which the effects of exposure-mediator and mediator-outcome had opposite signs. We also illustrated the case of figure 3C with the effects E→M and E→D as greater than zero and the effect M→D as less than zero in online supplementary B. More details of theoretical derivation can be found in online supplementary appendix.

### Scenario 2: two series mediators

Figure 1B is a depiction through two series mediators, decomposing total effects into direct effect (E→D) and indirect effect (E→M₁→M₂→D). The bias of varying across the effect of E→M₁ was greater than that of varying across the effect of M₂→D under adjustment for M₁, M₂ and M₁ M₂ together in figure 4, respectively. In this situation, the correlation of series mediators was strong enough to prevent M₂ from becoming an independent cause of the outcome.

### Scenario 3: two independent parallel mediators

Figure 1C shows that the exposure E independently causes M₁ and M₂ and indirectly influences the outcome

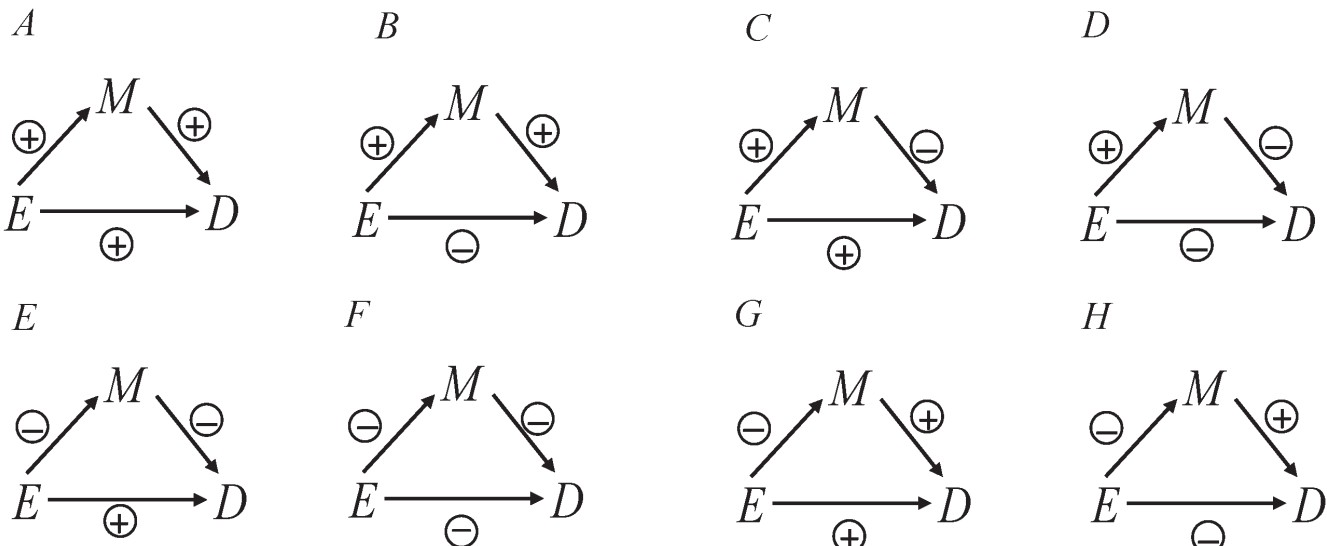

**Figure 3** Illustrating the use of positive and negative signs on edges E→M, M→D and E→D.

D through $M_1$ and $M_2$, forming three causal paths E→D, E→$M_1$→D and E→$M_2$→D. For figure 5, the results indicated that the bias of varying across the effect of E→$M_1$ was considerably greater than that of varying across the effect of $M_1$→D under adjustment for $M_1$ in figure 5A. However, the bias of varying across the effect of E→$M_2$ was nearly equal to that of varying across the effect of $M_2$→D under the identical model of adjustment for $M_1$ in figure 5A. Then, a result similar to the one above can be obtained in figure 5B. That is, the bias of the effect of E→$M_1$ was nearly equal to the effect of $M_1$→D and the bias of the effect of E→$M_2$ was greater than the effect of

$M_2$→D. In addition, figure 5C indicated that biases of varying across the effects of E→$M_1$ and E→$M_2$ were obviously greater than those of varying across the effects of $M_1$→D and $M_2$→D while simultaneously adjusting for $M_1$ and $M_2$.

### Scenario 4: two correlated parallel mediators

There exist five paths from E to D: E→D, E→$M_1$→D, E→$M_2$→D, E→$M_1$←$M_2$→D and E→$M_2$→$M_1$→D. In particular, the path E→$M_1$←$M_2$→D is a blocked path, due to $M_1$ being a collider node. Figure 6A indicated that the bias of varying across the effect of E→$M_1$ was clearly greater than

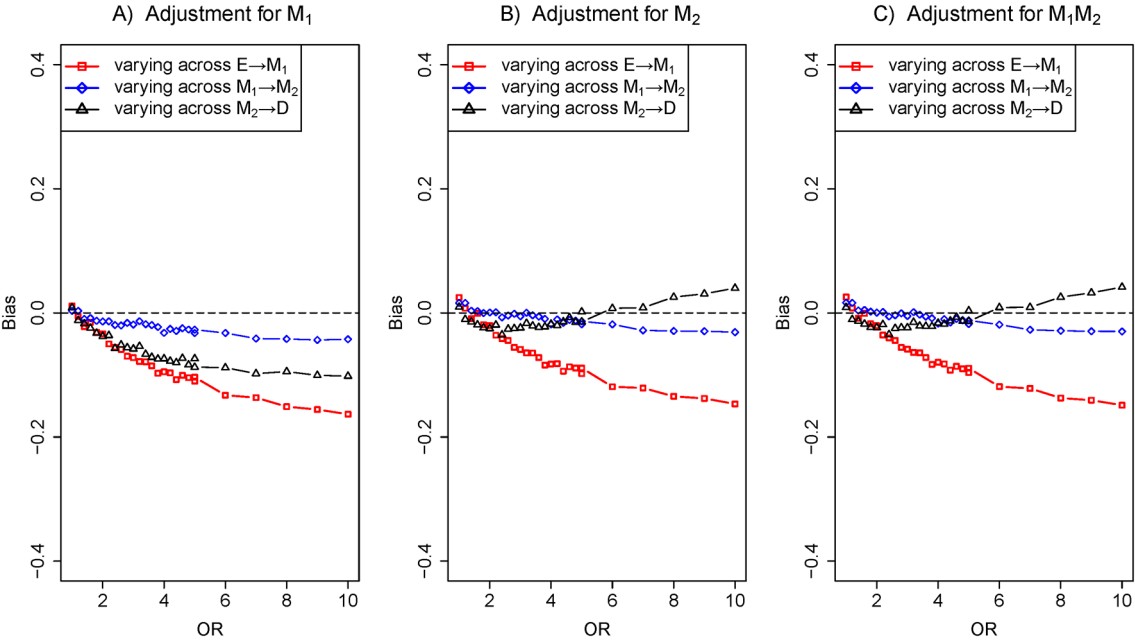

**Figure 4** The biases with the effects of E→$M_1$ (red), $M_1$→$M_2$ (blue) and $M_2$→D (black) increasing, respectively. Comparison of the bias of different effects in three adjustment models: (A) adjustment for $M_1$, (B) adjustment for $M_2$ and (C) adjustment for $M_1$ and $M_2$. The OR of target effect (eg, E→$M_1$) from 1 to 10 given the effect of $M_1$→$M_2$ fixed ln8 and other effects fixed ln2 (colour figure online).

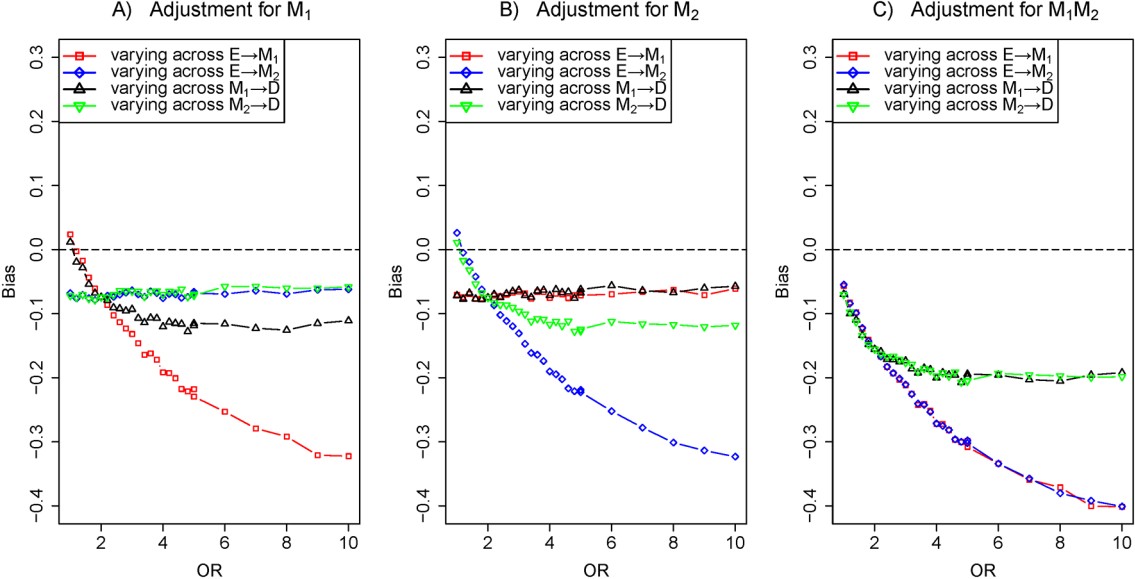

**Figure 5** The biases with the effects of E→M$_1$ (red), E→M$_2$ (blue), M$_1$→D (black) and M$_2$→D (green) increasing, respectively. Comparison of the bias of different effects in three adjustment models: (A) adjustment for M$_1$, (B) adjustment for M$_2$ and (C) adjustment for M$_1$ and M$_2$. The OR of target effects (eg, E→M$_1$) from 1 to 10 given other edges effects fixed ln2 (colour figure online).

that of varying across the effect of M$_1$→D under adjustment for M$_1$. However, the bias of varying across the effect of E→M$_2$ was almost equal to that of varying across the effect of M$_2$→D under the identical adjustment model. Similarly, an result of the behaviour of the biases is shown in figure 6B. That is, the bias of the effect of E→M$_1$ was nearly equal to the effect of M$_1$→D and the bias of the effect of E→M$_2$ was greater than the effect of M$_2$→D. In

addition, the biases of varying across the effects of E→M$_1$ and E→M$_2$ were greater than those of varying across the effects of M$_1$→D and M$_2$→D when adjusting for M$_1$ and M$_2$ in figure 6C. Simultaneously, the bias was more sensitive to the variation of the effect of E→M$_2$ than the effect of E→M$_1$ under adjustment for M$_1$ and M$_2$, while adjusting for the collider node M$_1$ would partially open the path E→M$_1$←M$_2$→D.

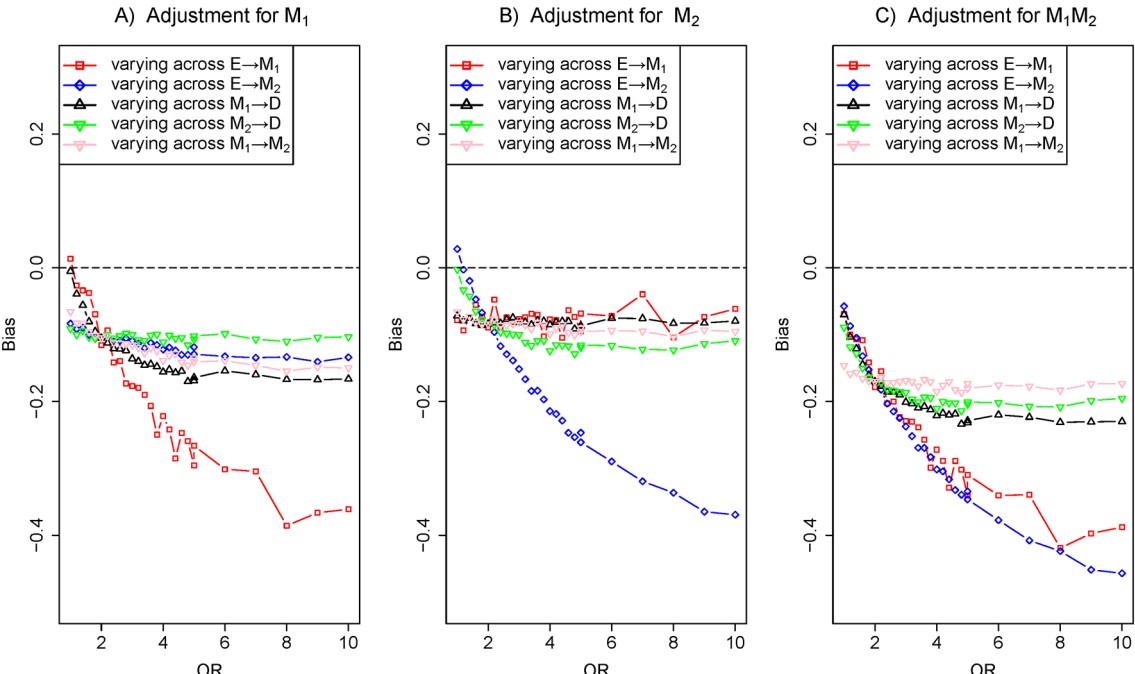

**Figure 6** The biases with the effects of E→M$_1$ (red), E→M$_2$ (blue), M$_1$→D (black), M$_2$→D (green) and the effect of M$_2$→M$_1$ (purple) increasing, respectively. Comparison of the bias of different effects in three adjustment models: (A) adjustment for M$_1$, (B) adjustment for M$_2$ and (C) adjustment for M$_1$ and M$_2$. The OR of target effects (eg, E→M$_1$) from 1 to 10 given other effects fixed ln2 (colour figure online).

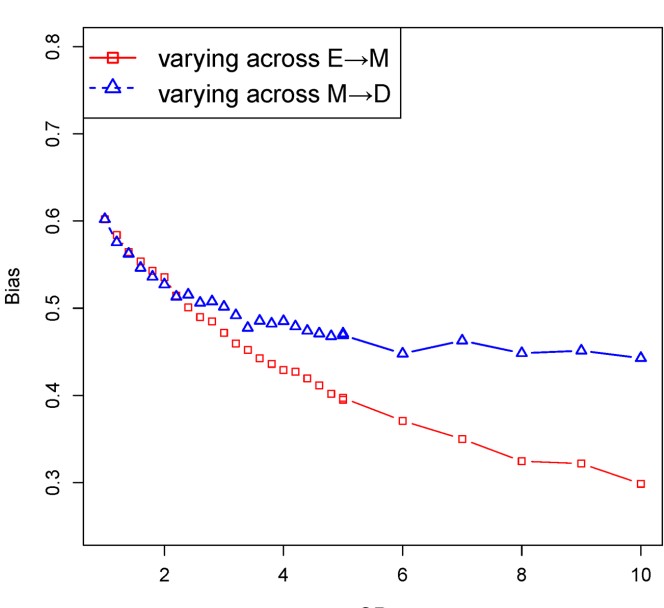

**Figure 7** The biases with the effects of E→M (red) and M→D (blue) increasing, respectively. Comparison of the bias of different effects in adjustment mediator M. The OR of target effects (eg, E→M) from 1 to 10 given the effects of causal edges fixed ln2 and the effect of confounder edges fixed ln5 (colour figure online).

### Scenario 5: a single mediator with an unobserved confounder

Figure 1E provides a causal diagram representing the relationship among exposure E, outcome D, mediator M and unobserved confounder U. It revealed that the bias of varying across the effect of E→M was lower than that of varying across the effect of M→D. An unobserved confounder distorted the association between the exposure and outcome (E←U→D) in figure 7.

### Scenario 6: two parallel mediators with an unobserved confounder

As described above, figure 1F is a depiction of two parallel mediators $M_1$ and $M_2$ with an unobserved confounder U. For figure 8, the bias of varying across the effect of E→$M_1$ was clearly less than that of varying across the effect of $M_1$→D under the adjustment for $M_1$ in figure 8A. However, the bias of varying across the effect of E→$M_2$ was greater than that of varying across the effect of $M_2$→D under the identical model adjusting for $M_1$. A similar result can also be obtained in figure 8B. In addition, biases of varying across the effects of E→$M_1$ and E→$M_2$ were distinctly less than those of varying across the effects of $M_1$→D and $M_2$→D under the common model of adjusting for $M_1$ and $M_2$ in figure 8C.

### APPLICATION

In this analysis, we evaluated two statistical models (unadjusted and M adjusted) to assess the effect of diabetes on cardiovascular diseases under scenario 1. Information from 22 900 individuals were collected from the Health Management Centre of Shandong Provincial Hospital. All individuals were urban Han Chinese and more than 20 years of age, and they underwent a physical examination in 2013. Many studies focused on the associations between diabetes and metabolic syndrome[34] and between metabolic syndrome and cardiovascular disease.[35]

The exposure indicator E takes a value of 1 if individuals suffer from diabetes and takes a value of 0 otherwise. The outcome D (cardiovascular diseases) takes a value of 1 if individuals are diagnosed with cardiovascular diseases and takes a value of 0 otherwise. The mediator M (metabolic syndrome) takes a value of 1 if individuals

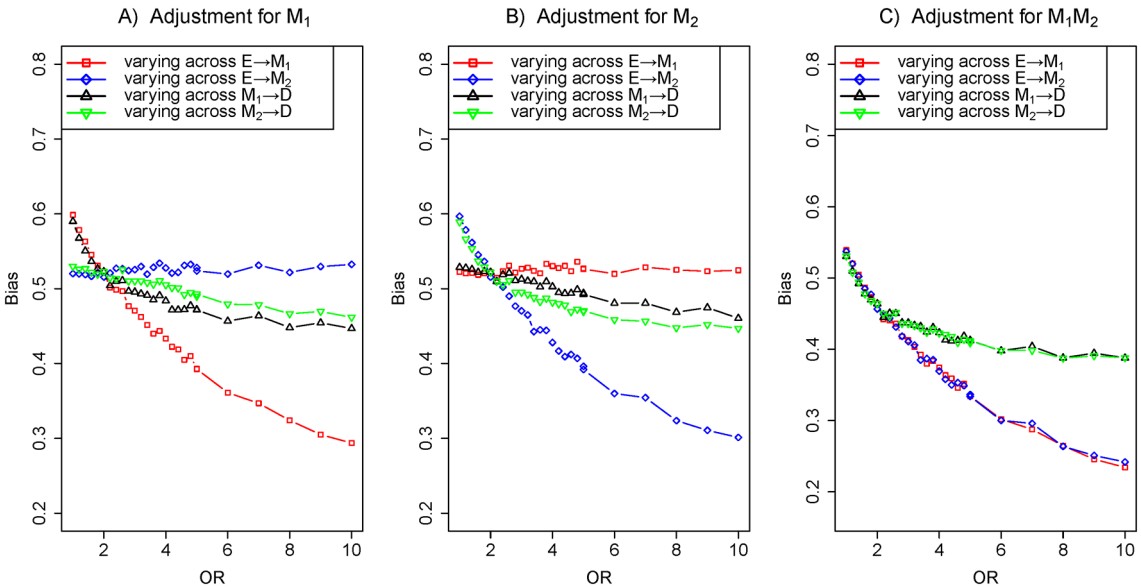

**Figure 8** The biases with the effects of E→$M_1$ (red), E→$M_2$ (blue), $M_1$→D (black) and $M_2$→D (green) increasing, respectively. Comparison of the bias of different effects in three adjustment models: (A) adjustment for $M_1$, (B) adjustment for $M_2$ and (C) adjustment for $M_1$ and $M_2$. The OR of target effects (eg, E→$M_1$) from 1 to 10 given the effects of causal edges fixed ln2 and the effect of confounder edges fixed ln5 (colour figure online).

diagnosed with metabolic syndrome and takes a value of 0 otherwise. After adjusting for age and gender, using the logistic regression model obtained the total effect of diabetes E on cardiovascular diseases D equal to β=0.598 (95% CI 0.307 to 0.877). Then, the effect of adjusting for metabolic syndrome M was equal to $\beta_M$=0.429 (95% CI 0.113 to 0.736). Therefore, the bias was, $\beta_M$−β=−0.169<0, suggesting that the effect of E on D was underestimated when adjusting for the mediator M. This bias can have negative implications on the interpretation of the effects of diabetes on cardiovascular diseases. The adjustment for the mediator produced biased estimates, and adjustment was thus inappropriate and should have been avoided. A specific example was the adjustment for time-varying confounders that are also mediators using methods including standardisation, inverse-probability weighting and G-estimation.[36] That is, investigators should remember to consider biological and clinical information when specifying a statistical model.

## DISCUSSION

In the paper, we dissected the sensitivity of bias to the variation of the effects of exposure-mediator and mediator-outcome when adjusting for mediators under the framework of the logistic regression model. In four scenarios (a single mediator in figure 1A of scenario 1, two series mediators in figure 1B of scenario 2, two independent parallel mediators in figure 1C of scenario 3 or two correlated parallel mediators in figure 1D of scenario 4), the bias of varying across the effect of exposure-mediator was greater than that of varying across the effect of mediator-outcome when adjusting for the mediator (figures 2, 4, 5 and 6). However, in two other scenarios (a single mediator or two independent parallel mediators in the presence of unobserved confounders in figure 1E of scenario 5 and figure 1F of scenario 6), the biases were more sensitive to the variation of the effect of mediator-outcome than the effect of exposure-mediator when adjusting for the mediator (figures 7 and 8).

Conditioning on a mediator is of concern in all areas of epidemiologic studies.[13 19 37] It indeed lead to bias in estimating the total effect of the exposure on the outcome.[8 22 23] Mediators and confounders are indistinguishable in terms of statistical association and conceptual grounds.[3] Most of the studies focus on the mediation effect analysis such as the calculation of direct effect and indirect effect.[20 21 38–41] Recently, some authors have used causal diagrams to describe how to appropriately handle matching variables. In addition, they have proven that matching on mediator M renders M and D independent (by design) in the matched study. Matching on variables that are affected by the exposure and the outcome, that is, mediators between the exposure and the outcome, would ordinary produce irremediable bias. Furthermore, matching on mediator M blocks the causal path E→M→D and thus produces unfaithfulness in estimating the total effect E on D.[31 42] Little effort has been made to learn the

performances of biases when adjusting for a mediator in estimating the total effect of an exposure on an outcome. Our study results revealed that the biases were more sensitive to the variation of the effects of exposure-mediator than effects of mediator-outcome when adjusting for the mediator in the absence of the unobserved confounder in causal diagrams (figure 1A–D). Nevertheless, for causal diagrams (figure 1E, F), the biases were more sensitive to the variation of effects of mediator-outcome than the effects of exposure-mediator when adjusting for a mediator in the presence of the unobserved confounder. Therefore, the biases of varying across different effects depended on the causal diagrams framework and whether an unobserved confounder existed.

The causal diagrams depicted in figure 1 are indeed very simplistic and concise, as they exclude the confounding factors of E and M as well as M and D. In practical applications, there exist some confounders in each pair of relationships among E, M and D. In addition, our simulation study was not comprehensive enough to evaluate the bias performances when adjusting for the mediator under logistic regression because it considered only binary variables, certain scenarios of effect size and common types of models. In medical research, regression modelling is commonly used to adjust for covariates associated with both the outcome and exposure. In this paper, the biases are defined by the difference between M-adjusted and unadjusted ORs, some of which is attributable to the non-collapsibility of the OR. In the field of causal inference, standardisation and inverse-probability weighting may obtain a different bias from that of regression modelling, and they may be better alternatives to calculate bias.[4 5] Therefore, in future research, the methods of standardisation and inverse-probability weighting could be used to calculate the biases of this paper definition. Future research should further reinforce the mechanisms and conceptual frameworks of confounders and mediators from causal diagrams to avoid falling into analytic pitfalls.

## CONCLUSION

In conclusion, the sensitivity of biases to the variation of the effects of exposure-mediator and mediator-outcome was related to whether there was an unobserved confounder in causal diagrams. The biases were more sensitive to the variation of the effects of exposure-mediator than the effects of mediator-outcome when adjusting for the mediator in the absence of unobserved confounders, while the biases were more sensitive to the variation of the effects of mediator-outcome than the effects of exposure-mediator in the presence of unobserved confounders.

**Acknowledgements** We would like to thank the anonymous reviewers and academic editors for providing us with constructive comments and suggestions and also wish to acknowledge our colleagues for their invaluable work.

**Contributors** TW and HL jointly conceived the idea behind the article and designed the study. TW conducted the literature review, performed the simulation and prepared the draft of the manuscript. PS, YY, XS, YL and ZY participated in

the design of the study and the revision of the manuscript. FX advised on critical revision of the manuscript for important intellectual content. All authors read and approved the final manuscript.

**Funding**   This work was supported by grants from the National Natural Science Foundation of China (grant numbers 81573259 and 81773547).

**Competing interests**   None declared.

**Ethics approval**   Ethics Committee of the School of Public Health (20140322), Shandong University. Written informed consent was obtained from all participants.

**Provenance and peer review**   Not commissioned; externally peer reviewed.

**Data sharing statement**   No additional data are available.

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
