## [Reviewer comments · BMJ Open]

ARTICLE DETAILS

TITLE (PROVISIONAL)	Sensitivity analysis for mistakenly adjusting for mediators in estimating total effect in observational studies
AUTHORS	Xue, Fuzhong; Wang, Tingting; Li, Hongkai; Su, Ping; Yu, Yuanyuan; Sun, Xiaoru; Liu, Yi; Yuan, Zhongshang

VERSION 1 - REVIEW

REVIEWER	Zhi Geng School of Mathematical Sciences, Peking University Beijing, China
REVIEW RETURNED	18-Jan-2017

GENERAL COMMENTS	The authors focused on six causal diagrams concerning different roles of mediators to compare the sensitivity of the effect of exposure-mediator with the effect of mediator-outcome in adjusting for mediator under the framework of logistic regression model. They reached some conclusions related to the sensitivity of different effects by some simulation studies. For some observational studies, it may be difficult to distinguish a cause variable and an effect variable of an exposure variable. They illustrate the biases in various cases induced by wrongly treating a mediator in the path from the exposure to the outcome as a confounder between the exposure and the outcome. The issue is important for observational studies, and the authors give some interesting results of simulation studies. Some comments and suggestions are given below. 1. For the six causal diagrams given in Fig. 1, the authors should try some theoretical results for evaluating the biases in the general cases. The authors show only simulation results for the cases of binary variables. It may be possible for the authors to derive a formula which expresses the total effect as a function of the effects $(E!M)$, $(M!D)$ and $(E!D)$. VanderWeele, T. J. (2010) Bias formulas for sensitivity analysis for direct and indirect effects. Epidemiology, Vol 21, No 4, 540-551. 2. For the diagram in Fig 1a, if we adjust for the mediator M, we obtain a direct effect of E on D (DE); if we do not adjust for M, we obtain the total effect of E
---

	on $D(TE)$. Thus the 'bias' may be the difference $TE - DE$, and it may be the indirect effect of E on D. For linear models, $D = a + b_E + c_M + \epsilon_D$, $M = \alpha + \beta_E + \epsilon_M$, we have $TE = b + c\beta$, $DE = b$, bias = $c - \beta$. What is that for the logistic models of the binary case? 3. For the six causal diagrams, it may be difficult to show the signs (positive, negative or null) of the biases by these simulation studies where the parameters are fixed to some specific values. It may be useful to use the signed DAGs for the discussion (VanderWeele and Robins, 2010, JRSS B, pp. 111-127). 4. Page 5 line 20, the definition of βM is not clear. Is it for a specific value M or for an expectation over M? Similarly for page 7, line 56. 5. Page 5, line 4, no definition of $Y(x)$. 6. Page 5, lines 9-33, the total causal effect is defined as β. The estimate $\hat{\beta M}$ should be denoted by $\hat{\beta M} = \text{logit}[\hat{P}(\dots)] - \text{logit}[\hat{P}(\dots)]$, where $\hat{P}(\dots)$ denotes the estimate of ... The bias should be defined by $E(\hat{\beta M}) - \beta$.
--	--

REVIEWER	Ilya Shpitser Johns Hopkins University, USA
REVIEW RETURNED	05-Feb-2017

GENERAL COMMENTS	In this paper, the authors using a simulation study to assess the extent of bias in causal effects if mediators were treated as confounders. This paper is a simulation study of a single parametric model, which I don't think is sufficiently novel to merit publication. Additional comments below. There appear to be some technical errors in this paper. --- The authors are not really comparing cases where M is treated as a mediator with cases where M is treated as confounder, as the title suggests. They are comparing cases where M is treated as mediator with cases where M is conditioned on at a particular value. Treating M as a confounder would imply using the adjustment formula at the bottom of p. 7, correct? That is the causal log(OR) would be $\log\left(\frac{p(D=1 do(E=1)) p(D=0 do(E=0))}{[p(D=0 do(E=1)) p(D=1 do(E=0))]}$
--

	 do(E=0)]), which does not equal what the authors have at the bottom of p. 7. Can the authors explain? --- equation on p. 5: can the authors explain why $p(D do(E))$ is equal to $\sum_M p(D E, M)$ in Fig. 1a? It's not even clear this is a normalized probability. What is the do-calculus derivation? This differs from the equation on p. 7. This does not sound right.
--	---

VERSION 1 – AUTHOR RESPONSE

Reviewer 1

Responses to Reviewer 1

Thank you very much for your insightful comments. Your comments have led to great improvement of our manuscript, and we have carefully revised the manuscript following your suggestions. Response to your specific comments are given below.

1. For the six causal diagrams given in Fig. 1, the authors should try some theoretical results for evaluating the biases in the general cases. The authors show only simulation results for the cases of binary variables. It may be possible for the authors to derive a formula which expresses the total effect as a function of the effects ($E \rightarrow M$), ($M \rightarrow D$) and ($E \rightarrow D$).

Response: Thanks for your insightful comments. The theoretical derivation for six causal diagrams are quite similar. Let D_e and M_e denote respectively the values of the outcome and mediator that would have been observed had the exposure E been set to level e . On the odds ratio ($OR_{E \rightarrow D}^{TE}$) scale, the total effect ($\beta_{E \rightarrow D}^{TE} = \log(OR_{E \rightarrow D}^{TE})$), comparing exposure level e with e^* , is given by

$$OR_{E \rightarrow D}^{TE} = \frac{P(D_e = 1) / \{1 - P(D_e = 1)\}}{P(D_{e^*} = 1) / \{1 - P(D_{e^*} = 1)\}}. \text{ While the effect } (\beta_{ED|M}(m)) \text{ of adjusting for mediator } M \text{ by}$$

logistic regression model can be given

$$\begin{aligned}\beta_{ED|M}(m) &= \text{logit}\{P(D=1|e=1,m)\} - \text{logit}\{P(D=1|e^*=0,m)\} \\ &= \log\left\{\frac{P(D=1|e=1,m)P(D=0|e^*=0,m)}{P(D=0|e=1,m)P(D=1|e^*=0,m)}\right\}\end{aligned}$$

Therefore, for the six causal diagrams, we could derive general formula of the bias by

$\beta_{ED|M}(m) - \beta_{E \rightarrow D}^{TE}$. However, the details of specific derivation are based on the diagrams structure.

We have added the details for Figure 1(a) as an example (please see our response to your comment 2).

As for expressing the total effect as a function of the effects ($E \rightarrow M$), ($M \rightarrow D$) and ($E \rightarrow D$), we take the Figure 1(a) as an example to illustrate the theoretical results. The total effect, comparing exposure level e with e^* , is given by

$$\begin{aligned}\beta_{E \rightarrow D}^{TE} &= \log(OR_{E \rightarrow D}^{TE}) = \log\left\{\frac{P(D_e=1)/\{1-P(D_e=1)\}}{P(D_{e^*}=1)/\{1-P(D_{e^*}=1)\}}\right\} \\ &= \log\left\{\frac{P(D_e=1) \times \{1-P(D_{e^*}=1)\}}{\{1-P(D_e=1)\} \times P(D_{e^*}=1)}\right\} \\ &= \log\left\{\frac{P(D=1|e=1) \times P(D=0|e^*=0)}{P(D=0|e=1) \times P(D=1|e^*=0)}\right\} \\ &= \log\left\{\frac{[\sum_m P(D=1|e=1,m)P(m|e=1)] \times [\sum_m P(D=0|e^*=0,m)P(m|e^*=0)]}{[\sum_m P(D=0|e=1,m)P(m|e=1)] \times [\sum_m P(D=1|e^*=0,m)P(m|e^*=0)]}\right\}\end{aligned}$$

The theoretical results of other causal diagrams in Figure 1 have been shown in the supplementary A.

2. For the diagram in Fig 1a, if we adjust for the mediator M , we obtain a direct effect of E on D (DE); if we do not adjust for M , we obtain the total effect of E on D (TE). Thus the 'bias' may be the difference $TE - DE$, and it may be the indirect effect of E on D . For linear models,

$$D = a + b^*E + c^*M + \varepsilon_D,$$

$$M = \alpha + \beta^*E + \varepsilon_M,$$

we have

$$TE = b + c^* \beta,$$

$$DE = b,$$

$$\text{bias} = c^* \beta.$$

What is that for the logistic models of the binary case?

Response: Thanks for your suggestions. For the diagram in Fig. 1(a), let D_e and M_e denote respectively the values of the outcome and mediator that would have been observed had the exposure E been set to level e . The logistic regression is,

$$\text{logit}\{P(D=1|m,e)\} = \alpha_1 + \beta_0 e + \beta_2 m.$$

The total effect ($\beta_{E \rightarrow D}^{TE}$) of E on D on the scale of logarithm odds ratio was equal to

$$\begin{aligned} \beta_{E \rightarrow D}^{TE} &= \log(OR_{E \rightarrow D}^{TE}) = \log \left\{ \frac{P(D_e=1)/\{1-P(D_e=1)\}}{P(D_{e^*}=1)/\{1-P(D_{e^*}=1)\}} \right\} \\ &= \log \left\{ \frac{P(D_e=1) \times \{1-P(D_{e^*}=1)\}}{\{1-P(D_e=1)\} \times P(D_{e^*}=1)} \right\} \\ &= \log \left\{ \frac{P(D=1|e=1) \times P(D=0|e^*=0)}{P(D=0|e=1) \times P(D=1|e^*=0)} \right\} \\ &= \log \left\{ \frac{[\sum_m P(D=1|e=1,m)P(m|e=1)] \times [\sum_m P(D=0|e^*=0,m)P(m|e^*=0)]}{[\sum_m P(D=0|e=1,m)P(m|e=1)] \times [\sum_m P(D=1|e^*=0,m)P(m|e^*=0)]} \right\} \end{aligned}$$

The effect ($\beta_{ED|M}(m)$) of adjusting for mediator M by logistic regression model can be given

$$\begin{aligned} \beta_{ED|M}(m) &= \text{logit}\{P(D=1|e=1,m)\} - \text{logit}\{P(D=1|e^*=0,m)\} \\ &= \log \left\{ \frac{P(D=1|e=1,m) \times \{1-P(D=1|e^*=0,m)\}}{\{1-P(D=1|e=1,m)\} \times P(D=1|e^*=0,m)} \right\} \\ &= \beta_0 \end{aligned}$$

β_0 denotes coefficient of the E adjusting for M using logistic regression model. Furthermore, the effect of adjusting for M was equal to the controlled direct effect. [VanderWeele and Valeri 2013] Therefore, the bias of adjusting for mediator using logistic regression model could be obtained i.e.

$\text{bias} = \beta_{ED|M}(m) - \beta_{E \rightarrow D}^{TE}$. Furthermore, we obtained the results of the bias when the effects $E \rightarrow M$, $M \rightarrow D$ and $E \rightarrow D$ were positive or negative (see page 8-9), and more details of theoretical derivation have been put in Appendix.

Valeri L, VanderWeele TJ. Mediation analysis allowing for exposure-mediator interactions and causal interpretation: theoretical assumptions and implementation with SAS and SPSS macros [J]. *Psychological methods*, 2013, 18(2): 137.

3. For the six causal diagrams, it may be difficult to show the signs (positive, negative or null) of the biases by these simulation studies where the parameters are fixed to some specific values. It may be useful to use the signed DAGs for the discussion (VanderWeele and Robins, 2010, JRSS B, pp. 111-127).

Response: Thank for pointing this out. Indeed, the simulation studies can obtain different signed biases (positive, negative or null) for the signed DAGs. In our manuscript, we aims to dissect the sensitivity of bias to the variation of the effects of exposure-mediator and mediator-outcome. According to your comments, for Figure 1a, we have adopted the signs DAGs that can be added to the edges of the directed acyclic graph to indicate the presence of a particular positive or negative effect in the Figure 3. The results illustrated that the bias was less than zero under the case of the effects of exposure-mediator and mediator-outcome sharing the same sign; the bias was greater than zero under circumstances of the effects of exposure-mediator and mediator-outcome having opposite signs. We have also showed the biases of the Figure S1 in supplementary B.

4. Page 5 line 20, the definition of β_M is not clear. Is it for a specific value M or for an expectation over M? Similarly for page 7, line 56.

Response: Sorry for bring you this confusion. The definition of β_M denoted the effect ($\beta_{ED|M}(m)$) of adjusting for mediator M by logistic regression model. The effect is given by

$$\begin{aligned} \beta_{ED|M}(m) &= \text{logit} \{P(D = 1 | e = 1, m)\} - \text{logit} \{P(D = 1 | e^* = 0, m)\} \\ &= \log \left\{ \frac{P(D = 1 | e = 1, m)P(D = 0 | e^* = 0, m)}{P(D = 0 | e = 1, m)P(D = 1 | e^* = 0, m)} \right\} \end{aligned}$$

where $P(D = 1 | e, m)$ denotes the probability of $D = 1$ when the exposure E , and mediator M , have been set to level e , and m , respectively. We have modified its definition in the revision to make it more clear (see page 5, line 16-20).

5. Page 5, line 4, no definition of $Y(x)$.

Response: Thanks for your attention. The definition of $Y(x)$, namely $P(y | do(x))$, stands for the probability of $Y = y$ when the exposure X has been set to level x . We have added these details into the revised manuscript (see page 5, line 5-7)

6. Page 5, lines 9-33, the total causal effect is defined as β .

The estimate β_M should be denoted by

$$\hat{\beta}_M = \text{logit}[\hat{P}(\dots)] - \text{logit}[\hat{P}(\dots)]$$

where $\hat{P}()$ denotes the estimate of ...

The bias should be defined by $E(\hat{\beta}_M) - \beta$.

Response : Thanks for pointing this out. According to your constructive comments, the total effect of exposure E on outcome D is defined as $\beta_{E \rightarrow D}^{TE}$. The effect estimation ($\hat{\beta}_{ED|M}(m)$) of adjusting for mediator M by logistic regression model was equal to

$$\hat{\beta}_{ED|M}(m) = \hat{\beta}_0 = \text{logit} \left\{ \hat{P}(D=1 | e=1, m) \right\} - \text{logit} \left\{ \hat{P}(D=1 | e^*=0, m) \right\}$$

We have redefined the bias by $E(\hat{\beta}_{ED|M}(m)) - \beta_{E \rightarrow D}^{TE}$.

Where $\hat{P}(D=1 | e=1, m)$ denotes the probability of $D=1$ when the exposure E , and mediator M , have been set to level $e=1$, and m , respectively. And $\hat{P}(D=1 | e^*=0, m)$ denotes the probability of $D=1$ when the exposure E , and mediator M , have been set to level $e^*=0$, and m , respectively. We have revised it in manuscript (see page 6, line 4-13).

Reviewer 2

Responses to Reviewer

We sincerely appreciate your effort in reviewing our manuscript. Your comments are very constructive and we have revised it accordingly. Responses to your specific comments are given below.

1. The authors are not really comparing cases where M is treated as a mediator with cases where M is treated as confounder, as the title suggests. They are comparing cases where M is treated as mediator with cases where M is conditioned on at a particular value.

Response: Thanks for pointing this out. In our manuscript, we focused on the consequences by wrongly adjusting for the mediator in estimating the total effect of the exposure on the outcome. To avoid the confusion you mentioned, we have modified the title as 'Sensitivity analysis for mistakenly adjusting for mediators in estimating total effect from the perspective of causal diagrams'. We mainly aim to compare biases of distinct adjustment strategies with and without adjusting for mediators, and

assessed the sensitivity of bias to variation of the effects of exposure-mediator and mediator-outcome.

2. Treating M as a confounder would imply using the adjustment formula at the bottom of p. 7, correct? That is the causal $\log(OR)$ would be

$\log([P(D=1 | do(E=1))P(D=0 | do(E=0))]/[P(D=0 | do(E=1))P(D=1 | do(E=0))])$,

which does not equal what the authors have at the bottom of p. 7.

Can the authors explain?

Response: Thank you very much for your insightful comments. In this manuscript, we aim to compare biases of distinct adjustment strategies with and without adjusting for mediators, not treating M as a confounder. Furthermore, we assessed the sensitivity of bias to variation of the effects of exposure-mediator and mediator-outcome. At the bottom of p.7 in our original manuscript, it means the effect (

$\beta_{ED|M}(m)$) of adjusting for mediator M by logistic regression model

$$\begin{aligned} \beta_{ED|M}(m) &= \text{logit} \{ P(D=1 | e=1, m) \} - \text{logit} \{ P(D=1 | e^*=0, m) \} \\ &= \log \left\{ \frac{P(D=1 | e=1, m)P(D=0 | e^*=0, m)}{P(D=0 | e=1, m)P(D=1 | e^*=0, m)} \right\} \end{aligned}$$

where $P(D=1 | e, m)$ denotes the probability of $D=1$ when the exposure E , and mediator M , have been set to level e , and m , respectively. We have added these detailed explanations in the revision to make it more clear (see page 5, line 16-20)

3. equation on p. 5:

can the authors explain why $P(D | do(E))$ is equal to $\sum_M P(D | do(E), M)$ in Fig. 1a? It's not

even clear this is a normalized probability. What is the do-calculus derivation? This differs from the equation on p. 7.

This does not sound right.

Response: Thanks for your attention. We have checked the equation again and found that we have made a mistake. The formula was

$$\begin{aligned}
\beta_{E \rightarrow D}^{TE} &= \log(OR_{E \rightarrow D}^{TE}) \\
&= \log \left\{ \frac{P(D_e = 1) / \{1 - P(D_e = 1)\}}{P(D_{e^*} = 1) / \{1 - P(D_{e^*} = 1)\}} \right\} \\
&= \text{logit} \{P(D_e = 1)\} - \text{logit} \{P(D_{e^*} = 1)\} \\
&= \text{logit} \{P(D = 1 | e = 1)\} - \text{logit} \{P(D = 1 | e^* = 0)\} \\
&= \text{logit} \left\{ \sum_m P(D = 1 | e = 1, m) P(m | e = 1) \right\} - \text{logit} \left\{ \sum_m P(D = 1 | e^* = 0, m) P(m | e^* = 0) \right\}
\end{aligned}$$

And we have corrected the equation in manuscript. (see page 6, line 3) The equation in p.5 was kept same with the equation in p.7.

The do-calculus derivation was based on the rules 2 of do calculus. Let X , Y and Z be arbitrary disjoint sets of nodes in a causal directed acyclic graph G .

$$P(y | do(x), do(z), w) = P(y | do(x), z, w) \quad \text{if } (Y \perp Z | X, W)_{G_{\bar{x}\bar{z}}}$$

This rule provides a condition for an external intervention $do(Z = z)$ to have the same effect on Y as the passive observation $Z = z$. The condition amounts to $\{X \cup W\}$ blocking all back-door paths from Z to Y (in $G_{\bar{x}}$), since $G_{\bar{x}\bar{z}}$ retains all (and only) such paths. $G_{\bar{x}}$ the graph obtained by deleting from G all arrows pointing to nodes in X . Likewise, $G_{\bar{x}}$ the graph obtained by deleting from G all arrows emerging from nodes in X . $G_{\bar{x}\bar{z}}$ denote the deletion of both incoming and outgoing arrows. (Pearl J. Causality[M]. Cambridge university press, 2000;85-86)

Therefore, in this paper, an external intervention $do(E = e)$ have same effect on the outcome D as the observation $E=e$, if there are no back-door paths from E to D in the corresponding causal DAGs.

$$\begin{aligned}
\beta_{E \rightarrow D}^{TE} &= \text{logit}\{P(D = 1 | do(e = 1))\} - \text{logit}\{P(D = 1 | do(e^* = 0))\} \\
&= \text{logit}\{P(D = 1 | e = 1)\} - \text{logit}\{P(D = 1 | e^* = 0)\} \\
&= \text{logit} \left\{ \sum_m P(D = 1 | e = 1, m) P(m | e = 1) \right\} - \text{logit} \left\{ \sum_m P(D = 1 | e^* = 0, m) P(m | e^* = 0) \right\} \\
&= \log \left\{ \frac{[\sum_m P(D = 1 | e = 1, m) P(m | e = 1)] \times [\sum_m P(D = 0 | e^* = 0, m) P(m | e^* = 0)]}{[\sum_m P(D = 0 | e = 1, m) P(m | e = 1)] \times [\sum_m P(D = 1 | e^* = 0, m) P(m | e^* = 0)]} \right\}
\end{aligned}$$

VERSION 2 – REVIEW

REVIEWER	Mohammad Ali Mansournia Department of Epidemiology and Biostatistics, School of Public Health, Tehran University of Medical Sciences, Iran
REVIEW RETURNED	03-May-2017

GENERAL COMMENTS

I have some important comments/suggestions and relevant citations for authors to consider:

1) On P. 5, the paper presents standardization to define the causal effect, but later it uses conditioning via regression modeling to adjust for the mediator. Although researchers often use regression modeling to adjust for covariates, some of the difference between M-adjusted and unadjusted ORs, called bias in the paper, is attributable to the non-collapsibility of OR (and it can be substantial unless the outcome is very rare in all E-M strata); see my next point below. I suggest that the authors at least mention standardization and inverse-probability weighting as better alternatives to calculate bias and cite the following relevant papers:

i) Mansournia MA, Altman DG. Inverse probability weighting. *BMJ*. 2016 Jan 15;352:i189.

ii) Gharibzadeh S, Mohammad K, Rahimiforoushani A, Amouzegar A, Mansournia MA. Standardization as a Tool for Causal Inference in Medical Research. *Arch Iran Med*. 2016 Sep;19(9):666-70.

2) Related to the first point, the odds ratio is not collapsible, so that the conditional odds ratio may differ from marginal odds ratio even if M is independent of E. On Page 8, lines 8-10, the authors describe this as "positive bias", though this is not really bias, but a consequence of non-collapsibility of odds ratio. In particular, if M is associated with D conditional on E and unconditionally unassociated with E, the M-conditional ORs must be farther from 1 than the unconditional OR (see reference (iii) below). However, both conditional and unconditional OR estimates are unbiased (technically consistent) for their population parameters. In this case, marginal OR obtained from standardization and inverse-probability weighting equals unconditional OR (total effect). I suggest that the authors correct their interpretation citing these two relevant papers for details:

iii) Mansournia MA, Hernán MA, Greenland S. Matched designs and causal diagrams. *Int J Epidemiol*. 2013;42(3):860-869.

iv) Mansournia MA, Greenland S. The relation of collapsibility and confounding to faithfulness and stability. *Epidemiology*. 2015;26(4):466-72.

3) I strongly suggest that the authors provide a real example in which the researchers may mistakenly include the mediators in the regression model. A particular example worth mentioning is adjustment for time-varying confounders which are also mediators as described in the following paper:

v) Mansournia MA, Danaei G, Forouzanfar MH, Mahmoudi M, Jamali M, Mansournia N, Mohammad K. Effect of physical activity on functional performance and knee pain in patients with osteoarthritis : analysis with marginal structural models. *Epidemiology*. 2012;23(4):631-40.

Of course, there would be no bias, if one adjust for time-varying

	confounders which are also mediators using G-methods including standardization, inverse-probability weighting, and G-estimation as mentioned in the paper (v), but there would be a tradeoff with traditional adjustment of regression modeling (a similar tradeoff for adjustment of confounders which are also colliders has been discussed in Greenland 2003 Epidemiology paper). 4) It should be mentioned as a limitation that the causal diagrams in Figure 1 are very simplistic as they all exclude common causes of E and M as well as common causes of M and D. In practice, all pairs of E, M, and D have some common causes. 5) The authors have described the consequence of mistakenly conditioning on mediators. It is worth mentioning that a similar context is inadvertently matching on mediators which biases both unconditional and conditional OR estimates as described in the Mansournia et al 2013 IJE paper; see reference (iii) above.
--	--

VERSION 2 – AUTHOR RESPONSE

Reviewer 3

Responses to Reviewer 3

Thank you very much for your insightful comments. Your comments have led to great improvement of our manuscript, and we have carefully revised the manuscript following your suggestions. Response to your specific comments are given below.

1) On P. 5, the paper presents standardization to define the causal effect, but later it uses conditioning via regression modeling to adjust for the mediator. Although researchers often use regression modeling to adjust for covariates, some of the difference between M-adjusted and unadjusted ORs, called bias in the paper, is attributable to the non-collapsibility of OR (and it can be substantial unless the outcome is very rare in all E-M strata); see my next point below. I suggest that the authors at least mention standardization and inverse-probability weighting as better alternatives to calculate bias and cite the following relevant papers:

i) Mansournia MA, Altman DG. Inverse probability weighting. BMJ. 2016 Jan 15;352:i189.

ii) Gharibzadeh S, Mohammad K, Rahimiforushani A, Amouzegar A, Mansournia MA. Standardization as a Tool for Causal Inference in Medical Research. Arch Iran Med. 2016 Sep;19(9):666-70.

Thanks for your valuable comments. Actually, various strategies are used to eliminate confounding bias in non-randomized controlled studies. The conventional approaches contain multivariate regression, stratification, standardization and inverse-probability weighting, etc.⁴⁻⁵

In medical research, regression modeling is commonly used to adjust for covariates associated with both the outcome and exposure. In this paper, the biases are defined by the difference between M-adjusted and unadjusted ORs, some of which is attributable to the non-collapsibility of OR. In the field of causal inference, standardization and inverse-probability weighting may obtain the different bias comparing with the regression modeling, and they may be better alternatives to calculate bias⁴⁻⁵. Therefore, in future research, the methods of standardization and inverse-probability weighting could be used to calculate the biases of this paper definition. We have mentioned these methods in introduction and discussion sections and cited the above references you mentioned, in the highlighted part in page 3 line 8-10 and page 15 line 8-17.

2) Related to the first point, the odds ratio is not collapsible, so that the conditional odds ratio may differ from marginal odds ratio even if M is independent of E. On Page 8, lines 8-10, the authors describe this as "positive bias", though this is not really bias, but a consequence of non-collapsibility of odds ratio. In particular, if M is associated with D conditional on E and unconditionally unassociated with E, the M-conditional ORs must be farther from 1 than the unconditional OR(see reference (iii) below). However, both conditional and unconditional OR estimates are unbiased (technically consistent) for their population parameters. In this case, marginal OR obtained from standardization and inverse-probability weighting equals unconditional OR (total effect). I suggest that the authors correct their interpretation citing these two relevant papers for details:

iii) Mansournia MA, Hernán MA, Greenland S. Matched designs and causal diagrams. Int J Epidemiol. 2013;42(3):860-869.

iv) Mansournia MA, Greenland S. The relation of collapsibility and confounding to faithfulness and stability. Epidemiology. 2015;26(4):466-72.

Thanks for your suggestions. *M* was associated with *D* conditional on *E* and unconditionally independent with *E*, *M* became an independent risk factor of the outcome, adjusting for *M* obtained a positive "bias". Such bias was a consequence of non-collapsibility of odds ratio, and the M-conditional ORs must be far from 1 than the unconditional ORs.³¹⁻³² Actually, both adjustment and non-adjustment for *M* should yield unbiased causal effect estimates. Certainly, in this case, both marginal OR and conditional OR obtained from standardization and inverse-probability weighting were equals to total effect.³³ We have corrected the interpretation and cited the above references in the highlighted part in page 8 line 14-21.

3) I strongly suggest that the authors provide a real example in which the researchers may mistakenly include the mediators in the regression model. A particular example worth mentioning is adjustment for time-varying confounders which are also mediators as described in the following paper:

v) Mansournia MA, Danaei G, Forouzanfar MH, Mahmoudi M, Jamali M, Mansournia N, Mohammad K. Effect of physical activity on functional performance and knee pain in patients with osteoarthritis : analysis with marginal structural models. *Epidemiology*. 2012;23(4):631-40.

Of course, there would be no bias, if one adjust for time-varying confounders which are also mediators using G-methods including standardization, inverse-probability weighting, and G-estimation as mentioned in the paper (v), but there would be a tradeoff with traditional adjustment of regression modeling (a similar tradeoff for adjustment of confounders which are also colliders has been discussed in Greenland 2003 *Epidemiology* paper).

Thanks for your suggestions. We have provided a real example in which the researchers may mistakenly include the mediators in the regression model. When adjustment for age and gender by using the logistic regression model can obtain the total effect of diabetes E on cardiovascular diseases D being equal to $\beta = 0.598$ (95% confidence interval (CI), 0.307~0.877). Then the effect of adjusting for metabolic syndrome M was equal to $\beta_M = 0.429$ (95% confidence interval (CI), 0.113~0.736). Therefore, the bias was $\beta_M - \beta = -0.169 < 0$, suggesting that the effect of E on D was underestimated under adjusting for mediator M . This bias can have negative implication on the interpretation of effect of diabetes on cardiovascular. The adjustment for mediator produced biased estimates, and thus adjustment is inappropriate and should be avoided. A particular example was adjustment for time-varying confounders which are also mediators using methods including standardization, inverse-probability weighting, and G-estimation.³⁶ That is to say, investigators should remember to consider biology and clinical information when specifying a statistical model. (see page 12-13)

4) It should be mentioned as a limitation that the causal diagrams in Figure 1 are very simplistic as they all exclude common causes of E and M as well as common causes of M and D. In practice, all pairs of E, M, and D have some common causes.

Thanks for pointing this out. The causal diagrams depicted in Figure 1 are indeed very simplistic and concise, as they all exclude confounding factors of E and M as well as M and D . In practical application, there exist some confounders in each pair of E , M , and D . Based on your comments, we have mentioned the limitation of the causal diagrams in Figure 1 in the section of discussion on page 15 line 3-5.

5) The authors have described the consequence of mistakenly conditioning on mediators. It is worth mentioning that a similar context is inadvertently matching on mediators which biases both unconditional and conditional OR estimates as described in the Mansournia et al 2013 IJE paper; see reference (iii) above.

Thanks for your comments.

Recently some authors used causal diagrams described how appropriate handling of the matching variables. And they have proved that matching on mediator M renders M and D independent (by design) in the matched study. Matching on variable that are affected by the exposure and the outcome, or mediators between the exposure and the outcome, would ordinary produce irremediable bias. Furthermore, matching on mediator M blocks the causal path $E \rightarrow M \rightarrow D$ and thus produces unfaithfulness for estimating the total effect E on D .^{31,42} We have modified and cited the above reference in the discussion section on page 14 line 10-16.

VERSION 3 – REVIEW

REVIEWER	Mohammad Ali Mansournia Department of Epidemiology and Biostatistics, School of Public Health, Tehran University of Medical Sciences, Tehran, Iran
REVIEW RETURNED	09-Jun-2017
GENERAL COMMENTS	The authors' revisions are quite responsive to my concerns and so I recommend acceptance.